# Genetic and Epigenetic Host–Virus Network to Investigate Pathogenesis and Identify Biomarkers for Drug Repurposing of Human Respiratory Syncytial Virus via Real-World Two-Side RNA-Seq Data: Systems Biology and Deep-Learning Approach

**DOI:** 10.3390/biomedicines11061531

**Published:** 2023-05-25

**Authors:** Bo-Wei Hsu, Bor-Sen Chen

**Affiliations:** Laboratory of Automatic Control, Signal Processing and Systems Biology, Department of Electrical Engineering, National Tsing Hua University, Hsinchu 30013, Taiwan; s110061613@m110.nthu.edu.tw

**Keywords:** human respiratory syncytial virus (hRSV), HPI-GWGEN, host–pathogen RNA-Seq data, KEGG pathways, DNN-based DTI model, drug design specification, system biology, multimolecule drug

## Abstract

Human respiratory syncytial virus (hRSV) affects more than 33 million people each year, but there are currently no effective drugs or vaccines approved. In this study, we first constructed a candidate host–pathogen interspecies genome-wide genetic and epigenetic network (HPI-GWGEN) via big-data mining. Then, we employed reversed dynamic methods via two-side host–pathogen RNA-seq time-profile data to prune false positives in candidate HPI-GWGEN to obtain the real HPI-GWGEN. With the aid of principal-network projection and the annotation of KEGG pathways, we can extract core signaling pathways during hRSV infection to investigate the pathogenic mechanism of hRSV infection and select the corresponding significant biomarkers as drug targets, i.e., TRAF6, STAT3, IRF3, TYK2, and MAVS. Finally, in order to discover potential molecular drugs, we trained a DNN-based DTI model by drug–target interaction databases to predict candidate molecular drugs for these drug targets. After screening these candidate molecular drugs by three drug design specifications simultaneously, i.e., regulation ability, sensitivity, and toxicity. We finally selected acitretin, RS-67333, and phenformin to combine as a potential multimolecule drug for the therapeutic treatment of hRSV infection.

## 1. Introduction

Human Respiratory Syncytial Virus (hRSV) is a negative-sense, single-stranded RNA virus that causes infections of the respiratory tract. More than 80% of children have been infected at least once by 2 years of age, and half of these children have had RSV twice [1]. In 2015, Li et al. have approximately estimated that 33.1 million RSV-associated lower respiratory tract infections (LRTI), resulted in about 3.2 million hospital admissions and 59,000 deaths in children younger than 5 years [2]. Thompson et al. have estimated that RSV accounts for approximately 10,000 deaths annually in elders who are over the age of 65 years in the US [3]. As stated above, we concluded that RSV not only infects children but also elders. In a recent study, it indicated that RSV infection accounted for 11 percent of hospitalizations for pneumonia, 11 percent for chronic obstructive pulmonary disease, 5 percent for congestive heart failure, and 7 percent for asthma [3]. Thus, RSV is not only a threat itself, but it can also develop other complications that are more threatening.

Currently, no powerful drug is available to prevent RSV. Although Ribavirin is the only drug approved by the Food and Drug Administration (FDA) for the treatment of RSV, it is very costly, teratogenic, and inconvenient [1]. In 1996, Randolph and Wang did a double-blind trial on hRSV. Although there were trends in the direction of benefit, the results showed that ribavirin does not significantly reduce the mortality rate and decrease respiratory deterioration [4]. For an RSV vaccine, there is no available vaccine approved by FDA. Although the FDA has approved four candidate vaccines, we still need a convenient and efficient treatment for RSV. According to the above analysis, a new RSV treatment is urgent for RSV-infected patients.

In PhRMA (Pharmaceutical Research and Manufacturers of America) Biopharmaceutical Research in 2015, it indicates that the average cost to research and develop each successful drug is estimated to be $2.6 billion. The overall probability of clinical success is estimated to be less than 12%. Thus, a lower time cost and money cost drug-discovery method is needed for the whole pharmaceutical industry. Given the above analysis, drug repurposing is becoming an attractive method because it lowers much of the time for drug clinical trials and the development cost simultaneously. As a famous example, Viagra and Thalidomide are designed to reduce pulmonary arterial hypertension and nervousness at first; after their clinical trials failed, the researchers accidentally found they can cure erectile dysfunction and leprosy [5].

Nowadays, more and more data can be accessed, e.g., in October 2016, the EMA (European Medicines Agency) started directly providing clinical-trial data submitted by pharmaceutical companies on six different drugs. We can reanalyze these for drug repurposing which may reveal new targets and pathways that can be further exploited for drugs [6]. The attitude of EMA shows that drug repurposing is a trend in the pharmaceutical industry. Although the performance of some drug repurposing is doubtful, drug repurposing is still a low-risk method for drug discovery.

In this study, we established a multistep drug-repurposing workflow which is shown in Figure 1 by system identification, principal host–pathogen network projection, core signaling pathway of hRSV infection, biomarker identification, and drug discovery by a deep neural network-based drug–target interaction (DTI) model and drug design specification. First of all, we constructed a candidate host–pathogen interspecies genome-wide genetic and epigenetic interaction network (HPI-GWGEN) by big-data mining from multiple host–pathogen regulation/interaction databases. Then, we trimmed candidate HPI-GWGEN into real HPI-GWGEN by system identification via two-side host–pathogen time-profile RNA-seq data. The system identification is employed to construct the real HPI-GWGEN by identifying real regulations and interactions between each gene, protein, and virus via two-side RNA-Seq time-profile data of the hRSV and host. Then we prune the false-positive interactions based on system model order by Akaike information criterion (AIC) which can identify the real number of interactions and regulations [7,8,9,10]. Since there are too many interactions and regulations between the nodes of proteins, genes, microRNA, lncRNA, and virus genes in HPI-GWGEN, therefore, we proposed the principal-network projection (PNP) method to calculate the projection value of each node on the principal-network structure (85%) of HPI-GWGEN to rank the significance of each node and to further extract the core HPI-GWGEN from real HPI-GWGEN based on their projection (significance) values. To understand the core signaling pathways of hRSV infection, we used DAVID (database for annotation, visualization, and integrated discovery) and the annotation of KEGG pathways to obtain the core HPI signaling pathways of hRSV infection. After obtaining the core HPI signaling pathways and their downstream cellular dysfunctions in the hRSV infection progression to further investigate the pathogenic mechanism of hRSV infection, consequently, we selected several significant biomarkers i.e., IRF3, STAT3, TRAF6, TYK2, and MAVS as the drug targets for therapeutic treatment of hRSV infection. Then, we designed a DNN (deep neural network) as drug–target interaction (DTI) model to be trained by the drug–target interaction from drug–target interaction databases. After the training of the DTI model, we could predict the candidate multimolecule drugs which interact with these significant biomarkers we selected. Finally, we could evaluate the potential multimolecule drug based on drug design specifications such as regulation ability, high sensitivity, adequate excretion, toxicity, and drug-likeness as a selection benchmark. After all evaluations of systematic drug selection, we purposed acitretin, RS-67333, and phenformin to construct a multimolecule drug as the clinical-trial recommendation for hRSV treatment.

## 2. Material and Methods

### 2.1. Overview of Systematic Drug Discovery for hRSV Infection via Systems-Biology Method

In order to investigate the pathogenic mechanism for hRSV infection, we tried to reverse the hRSV infection progress and construct the real HPI-GWGEN via two-side RNA-Seq data of hRSV and human A549 cells. Since A549 is the most widespread cell line and hRSV is a respiratory virus, it seems reasonable to use a lung cell line even though hRSV mostly infects respiratory epithelial cells first. In addition, the construction of the candidate HPI-GWGEN is obtained from various datasets and cell lines, so we need the system identification to prune false positives to let real HPI-GWGEN suitable for hRSV infection. The GenBank accession number of hRSV is AF035006, which is a recombinant mutant virus of subgroup A. Given that the real HPI-GWGEN was too complicated to annotate by KEGG pathways, we applied the PNP to rank the significance of all node proteins, genes, microRNA, lncRNA, and virus genes in real HPI-GWGEN and then extract the most significant nodes to consist of core HPI-GWGEN. With the annotation of KEGG pathways, we could construct the core HPI signaling pathways of hRSV infection. From the downstream of cellular dysfunctions and the upstream of regulations of core signaling pathways of hRSV infection, we could finally discover the significant biomarkers of the pathogenic mechanism as drug targets for the therapeutic treatment of hRSV infection. After finding the drug targets, we purposed a DNN-DTI model to help us find the potential drugs which target these biomarkers. The DNN-DTI model was trained by drug–target interaction databases and was used to predict the candidate drugs that could interact with target genes. Subsequently, we purposed three drug specifications, i.e., regulation ability, sensitivity, and toxicity on molecular drugs to select a multimolecule drug as the clinical trial recommendation for hRSV treatment. The flowchart of the whole systematic drug discovery procedure is shown in Figure 1.

### 2.2. Construction of Candidate HPI-GWGEN by Database Mining and Integration

The purpose of constructing candidate HPI-GWGEN by big-data mining from databases is to investigate the possible interactions and regulations among proteins and genes between the host and hRSV. In the candidate HPI-GWGEN, if any two genes and proteins may have a regulatory or interaction relationship, we will set it true, and then use the Boolean type to express the entire HPI-GWGEN.

The candidate HPI-GWGEN contains two networks, including the candidate HPI protein–protein interaction network (HPI-PPIN) and the candidate HPI gene-regulation network (HPI-GRN). The host intraspecies of candidate HPI-PPIN were constructed by the big-data mining of several databases, including the Database of Interacting Proteins (DIP) [6], the Biological General Repository for Interaction Datasets (BioGRID) [11], the Biomolecular Interaction Network Database (BIND) [12], IntAct [13], and the Molecular Interaction Database (MINT) [14]. The host intraspecies of candidate HPI-GRN include miRNA, lncRNA, and transcription factors regulation. The candidate host miRNA and lncRNA regulations were constructed by the big-data mining of the Target Scan Human database [15], CircuitsDB [16], and starBase v2.0 [17]; the candidate host transcription factors regulations were constructed by the big-data mining of ITFP [18], TRANSFAC [19], and HTRIdb [20]. Due to the lack of hRSV–host interactions and regulations database, we assumed that all the hRSV–host interactions and regulations in candidate HPI-GWGEN are true; then, we purposed a system order identification method [7,8] to estimate the true order by two-side host–pathogen RNA-Seq time-profile data and eliminate the false positives in candidate HPI-GWGEN to obtain real HPI-GWGEN.

### 2.3. HPI RNA-Seq Time-Profile Data of Human A549 Cell and hRSV

The HPI RNA-seq data of the human A549 cell and hRSV used in the study are from National Center for Biotechnology Information (GEO number: GSE55963). This dataset included 8 time points: 0, 2, 4, 8, 12, 16, 20, and 24 hpi (hour postinfection) on human A549 cell and hRSV. These two-side RNA-seq time-profile data were employed to identify the true system parameters of candidate HPI-GWGEN by system-identification methods. We also purposed Genecode v35/v27 annotation on this data; the genes were sorted into five types, including proteins, miRNA, lncRNA, receptors, and transcription factors.

### 2.4. Construction of Dynamic Model of HPI-GWGEN for hRSV Infection

For candidate HPI-PPIN, the dynamic-interaction model of the host proteins can be modeled as the following equation:(1)$$\begin{array}{l} p_{i}{}^{H}(t+1)=p_{i}{}^{H}(t)+\sum_{j=1}^{HH_{i}}I_{ij}{}^{HH}p_{i}{}^{H}(t)p_{j}{}^{H}(t)+\sum_{k=1}^{HP_{i}}I_{ik}{}^{HP}p_{i}{}^{H}(t)p_{k}{}^{P}(t)\\ \;\;\;\;\;+\alpha_{Hi}p_{i}{}^{H}(t)-\gamma_{Hi}p_{i}{}^{H}(t)+\beta_{Hi}+n_{Hi}(t)\\ \alpha_{Hi}\geq 0\;\mathrm{and}\;-\gamma_{Hi}\leq 0,\;\mathrm{for}\;i=1,\;2,\;\cdots ,\;H \end{array}$$
where $p_{i}{}^{H}(t)$, $p_{j}{}^{H}(t)$ indicate the expression level of the ith host protein, the jth host protein at time point t, respectively; $HH_{i}$ and $HP_{i}$ indicate the number of host proteins and pathogen proteins interacting with the ith host protein, respectively; $I_{ij}{}^{HH}$ and $I_{ik}{}^{HP}$ indicate the interaction ability between the ith host protein and the jth host protein and between the ith host protein and the kth pathogen protein, respectively; $\alpha_{Hi}$, $-\gamma_{Hi}$, $\beta_{Hi}$ indicate the translation rate, the degradation rate, and the basal activity level of the ith host protein, respectively; $n_{Hi}(t)$ is the stochastic noise of the ith host protein at time point t; H indicates the total number of host proteins. Basal activity level can model the unknown regulations or interactions, ex. methylation or phosphorylation.

For candidate HPI-PPIN, the dynamic-interaction model of the pathogen proteins can be modeled as the following equation:(2)$$\begin{array}{l} p_{i}{}^{P}(t+1)=p_{i}{}^{P}(t)+\sum_{j=1}^{PH_{i}}I_{ij}{}^{PH}p_{i}{}^{P}(t)p_{j}{}^{H}(t)+\sum_{k=1}^{PP_{i}}I_{ik}{}^{PP}p_{i}{}^{P}(t)p_{k}{}^{P}(t)\\ \;\;\;\;\;+\alpha_{Pi}p_{i}{}^{P}(t)-\gamma_{Pi}p_{i}{}^{P}(t)+\beta_{Pi}+n_{Pi}(t)\\ \alpha_{Pi}\geq 0\;\mathrm{and}\;-\gamma_{Pi}\leq 0,\;\mathrm{for}\;i=1,\;2,\;\cdots ,\;P \end{array}$$
where $p_{i}{}^{P}(t)$ and $p_{j}{}^{H}(t)$ indicate the expression level of the ith pathogen protein and the jth host protein at time point t, respectively; $PH_{i}$ and $PP_{i}$ indicate the number of host proteins and pathogen proteins interacting with the ith host pathogen, respectively; $I_{ij}{}^{PH}$ and $I_{ik}{}^{PP}$ indicate the interaction ability between the ith pathogen protein and the jth host protein and between the ith pathogen protein and the kth pathogen protein, respectively; $\alpha_{Pi}$, $-\gamma_{Pi}$, and $\beta_{Pi}$ indicate the translation rate, the degradation rate, and the basal activity level of the ith pathogen protein, respectively; $n_{Pi}(t)$ is the stochastic noise of the ith pathogen protein at time point t; P indicates the total number of pathogen proteins.

For candidate HPI-GRN, the dynamic-regulation models of the host genes, which including proteins, miRNAs, lncRNAs, transcription factors, and receptors, can be modeled by the following equations:(3)$$\begin{array}{l} g_{i}{}^{H}(t+1)=g_{i}{}^{H}(t)+\sum_{j=1}^{GT_{i}}R_{ij}{}^{GT}f_{j}{}^{H}(t)+\sum_{k=1}^{GM_{i}}R_{ik}{}^{GM}g_{i}{}^{H}(t)m_{k}{}^{H}(t)\\ \;\;\;\;\;+\sum_{q=1}^{GL_{i}}R_{iq}{}^{GL}l_{q}{}^{H}(t)-\gamma_{Gi}g_{i}{}^{H}(t)+\beta_{Gi}+n_{Gi}(t)\\ R_{ik}{}^{GM}\leq 0\;\mathrm{and}\;-\gamma_{Gi}\leq 0,\;\mathrm{for}\;i=1,\;2,\;\cdots ,{\;G}^{H} \end{array}$$
where $g_{i}{}^{H}(t)$, $f_{j}{}^{H}(t)$, $m_{k}{}^{H}(t)$, and $l_{q}{}^{H}(t)$ indicate the expression level of the ith host genes, the jth host transcription factors, the kth host miRNAs, and the kth host lncRNAs at time point t, respectively; $GT_{i}$, $GM_{i}$, and $GL_{i}$ indicate the number of host transcription factors, miRNA, and lncRNA interacting with ith host gene, respectively; $R_{ij}{}^{GT}$, $R_{ik}{}^{GM}$, and $R_{iq}{}^{GL}$ indicate the regulation ability of the ith host gene regulated by the jth host transcription factor, the kth host miRNA, and the qth host lncRNA respectively; $-\gamma_{Gi}$ and $\beta_{Gi}$ indicate the degradation rate and the basal activity level of the ith host gene, respectively; $n_{Gi}(t)$ is the stochastic noise of the ith host gene at time point t; G indicates the total number of host genes.

For candidate HPI-GRN, the dynamic-regulation models of the pathogen genes can be modeled as the following equations:(4)$$\begin{array}{l} g_{i}{}^{P}(t+1)=g_{i}{}^{P}(t)+\sum_{j=1}^{VT_{i}}R_{ij}{}^{VT}f_{j}{}^{H}(t)+\sum_{k=1}^{VM_{i}}R_{ik}{}^{VM}g_{i}{}^{P}(t)m_{k}{}^{H}(t)\\ \;\;\;\;\;+\sum_{q=1}^{VL_{i}}R_{iq}{}^{VL}l_{q}{}^{H}(t)+\sum_{z=1}^{VV_{i}}R_{iz}{}^{VV}g_{z}{}^{P}(t)-\gamma_{Vi}g_{i}{}^{P}(t)+\beta_{Vi}+n_{Vi}(t)\\ R_{ik}{}^{VM}\leq 0\;\mathrm{and}\;-\gamma_{Vi}\leq 0,\;\mathrm{for}\;i=1,\;2,\;\cdots ,{\;G}^{P} \end{array}$$
where $g_{i}{}^{P}(t)$, $f_{j}{}^{H}(t)$, $m_{k}{}^{H}(t)$, and $l_{q}{}^{H}(t)$ indicate the expression level of the ith pathogen gene, the jth host transcription factor, the kth host miRNA, and the kth host lncRNA at time point t, respectively; $VT_{i}$, $VM_{i}$, and $VL_{i}$ indicate the number of host transcription factors, miRNAs, and lncRNAs interacting with the ith pathogen (virus) gene, respectively; $R_{ij}{}^{VT}$, $R_{ik}{}^{VM}$, $R_{iq}{}^{VL}$, and $R_{iz}{}^{VV}$ indicate the regulation ability of the ith pathogen (virus) gene regulated by the jth host transcription factor, kth host miRNA, qth host lncRNA, and the zth pathogen gene respectively; $-\gamma_{Vi}$ and $\beta_{Vi}$ indicate the degradation rate and the basal activity level of the ith pathogen (virus) gene, respectively; $n_{Vi}(t)$ is the stochastic noise of the ith pathogen (virus) gene at time point t; V indicates the total number of pathogen (virus) genes.

### 2.5. Parameter Estimation of Dynamic Model for Candidate HPI-GWGEN by System Identification Method for hRSV Infection Progression

Since the databases are noisy, there are a large number of false positives in the candidate HPI-GWGEN. Based on the discrete-time dynamic model Equations (1)–(4), we purposed system identification based on the HPI RNA-seq time-profile data (GSE55963) to prune false-positive regulations and interactions in candidate HPI-GWGEN to obtain the real HPI-GWGEN during the hRSV infection. In our raw RNA-seq data, there are only 8 time points that may lead to the overfitting in the parameter estimation of the HPI-GWGEN, so we applied the cubic spline interpolation to expand our time points to prevent overfitting in the parameter estimation process. Then, we purposed the Akaike information criterion (AIC) to find the correct system order by the trade-off between the model complexity and least square parameter estimation error [7].

To estimate the parameters in Equations (1)–(4), we rearranged each equation in linear regression form as follows:(5)$$\begin{array}{ll} p_{i}{}^{H}(t+1)=\left[p_{i}{}^{H}(t)p_{1}{}^{H}(t)\;\cdots \;p_{i}{}^{H}(t)p_{Hi}{}^{H}(t)\;p_{i}{}^{H}(t)p_{1}{}^{P}(t)\;\cdots \;p_{i}{}^{H}(t)p_{Pi}{}^{P}(t)\;p_{i}{}^{H}(t)\;p_{i}{}^{H}(t)\;1\right] & \left[\begin{array}{c} I_{i1}{}^{HH}\\ \begin{array}{c} \vdots \\ \begin{array}{c} \begin{array}{c} I_{iH_{i}}{}^{HH}\\ I_{i1}{}^{HP}\\ \vdots \end{array}\\ I_{iP_{i}}{}^{HP}\\ \alpha_{Hi} \end{array}\\ 1-\gamma_{Hi} \end{array}\\ \beta_{Hi} \end{array}\right]+n_{Hi}(t)\\ \to p_{i}{}^{H}(t+1)=A_{Hi}(t)I_{Hi}+n_{Hi}(t),\;\mathrm{for}\;i=1,2,\cdots ,H \end{array}$$
(6)$$\begin{array}{ll} p_{i}{}^{P}(t+1)=\left[p_{i}{}^{P}(t)p_{1}{}^{H}(t)\;\cdots \;p_{i}{}^{P}(t)p_{Hi}{}^{H}(t)\;p_{i}{}^{P}(t)p_{1}{}^{P}(t)\;\cdots \;p_{i}{}^{P}(t)p_{Pi}{}^{P}(t)\;p_{i}{}^{P}(t)\;p_{i}{}^{P}(t)\;1\right] & \left[\begin{array}{c} I_{i1}{}^{PH}\\ \begin{array}{c} \vdots \\ \begin{array}{c} \begin{array}{c} I_{iH_{i}}{}^{PH}\\ I_{i1}{}^{PP}\\ \vdots \end{array}\\ I_{iP_{i}}{}^{PP}\\ \alpha_{Pi} \end{array}\\ 1-\gamma_{Pi} \end{array}\\ \beta_{Pi} \end{array}\right]+n_{Pi}(t)\\ \to p_{i}{}^{P}(t+1)=A_{Pi}(t)I_{Pi}+n_{Pi}(t),\;\mathrm{for}\;i=1,2,\cdots ,P \end{array}$$
(7)$$\begin{array}{ll} g_{i}{}^{H}(t+1)=\left[g_{i}{}^{H}(t)m_{1}{}^{H}(t)\;\cdots \;g_{i}{}^{H}(t)m_{M}{}^{H}(t)\;f_{1}{}^{H}(t)\;\cdots \;f_{F}{}^{H}(t)\;l_{1}{}^{H}(t)\;\cdots \;l_{L}{}^{H}(t)\;g_{i}{}^{H}(t)\;1\right] & \left[\begin{array}{c} R_{i1}{}^{GM}\\ \vdots \\ R_{iM}{}^{GM}\\ R_{i1}{}^{GT}\\ \vdots \\ R_{iF}{}^{GT}\\ R_{i1}{}^{GL}\\ \vdots \\ R_{iL}{}^{GL}\\ 1-\gamma_{Gi}\\ \beta_{Gi} \end{array}\right]+n_{Gi}(t)\\ \to g_{i}{}^{H}(t+1){=A}_{Gi}(t)R_{Gi}+n_{Gi}(t),\;\mathrm{for}\;i=1,2,\cdots ,G^{H} \end{array}$$
(8)$$\begin{array}{ll} g_{i}{}^{P}(t+1)=\left[g_{i}{}^{P}(t)m_{1}{}^{H}(t)\;\cdots \;g_{i}{}^{P}(t)m_{M}{}^{H}(t)\;f_{1}{}^{H}(t)\;\cdots \;f_{F}{}^{H}(t)\;l_{1}{}^{H}(t)\;\cdots \;l_{L}{}^{H}(t)\;g_{1}{}^{P}(t)\;\cdots \;g_{V}{}^{P}(t)g_{i}{}^{P}(t)\;1\right] & \left[\begin{array}{c} R_{i1}{}^{VM}\\ \vdots \\ R_{iM}{}^{VM}\\ R_{i1}{}^{VT}\\ \vdots \\ R_{iF}{}^{VT}\\ R_{i1}{}^{VL}\\ \vdots \\ R_{iL}{}^{VL}\\ \begin{array}{c} \begin{array}{c} R_{i1}{}^{VV}\\ \vdots \end{array}\\ R_{iV}{}^{VV}\\ 1-\gamma_{Vi} \end{array}\\ \beta_{Vi} \end{array}\right]+n_{Vi}(t)\\ \to g_{i}{}^{P}(t+1){=A}_{Vi}(t)R_{Vi}+n_{Vi}(t),\;\mathrm{for}\;i=1,2,\cdots ,G^{P} \end{array}$$

Then, we could transform Equations (5)–(8) to the following augmented regression equations, respectively. *T* indicates the total number of time points of HPI RNA-seq time-profile data after interpolation.
(9)$$\begin{array}{l} \left[\begin{array}{c} p_{i}{}^{H}(t_{2})\\ p_{i}{}^{H}(t_{3})\\ \vdots \\ p_{i}{}^{H}(t_{T}) \end{array}\right]=\left[\begin{array}{c} A_{Hi}(t_{2})\\ A_{Hi}(t_{3})\\ \vdots \\ A_{Hi}(t_{T}) \end{array}\right]I_{Hi}+\left[\begin{array}{c} n_{Hi}(t_{2})\\ n_{Hi}(t_{3})\\ \vdots \\ n_{Hi}(t_{T}) \end{array}\right]\\ \to P_{Hi}=\phi_{Hi}I_{Hi}+\varepsilon_{Hi},\;\mathrm{for}\;i=1,2,\cdots ,H \end{array}$$
(10)$$\begin{array}{l} \left[\begin{array}{c} p_{i}{}^{P}(t_{2})\\ p_{i}{}^{P}(t_{3})\\ \vdots \\ p_{i}{}^{P}(t_{T}) \end{array}\right]=\left[\begin{array}{c} A_{Pi}(t_{2})\\ A_{Pi}(t_{3})\\ \vdots \\ A_{Pi}(t_{T}) \end{array}\right]I_{Hi}+\left[\begin{array}{c} n_{Pi}(t_{2})\\ n_{Pi}(t_{3})\\ \vdots \\ n_{Pi}(t_{T}) \end{array}\right]\\ \to P_{Pi}=\phi_{Pi}I_{Pi}+\varepsilon_{Pi},\;\mathrm{for}\;i=1,2,\cdots ,P \end{array}$$
(11)$$\begin{array}{l} \left[\begin{array}{c} g_{i}{}^{H}(t_{2})\\ g_{i}{}^{H}(t_{3})\\ \vdots \\ g_{i}{}^{H}(t_{T}) \end{array}\right]=\left[\begin{array}{c} A_{Gi}(t_{2})\\ A_{Gi}(t_{3})\\ \vdots \\ A_{Gi}(t_{T}) \end{array}\right]I_{Hi}+\left[\begin{array}{c} n_{Gi}(t_{2})\\ n_{Gi}(t_{3})\\ \vdots \\ n_{Gi}(t_{T}) \end{array}\right]\\ \to G_{Hi}=\phi_{Gi}R_{Gi}+\varepsilon_{Gi},\;\mathrm{for}\;i=1,2,\cdots ,G^{H} \end{array}$$
(12)$$\begin{array}{l} \left[\begin{array}{c} g_{i}{}^{P}(t_{2})\\ g_{i}{}^{P}(t_{3})\\ \vdots \\ g_{i}{}^{P}(t_{T}) \end{array}\right]=\left[\begin{array}{c} A_{Vi}(t_{2})\\ A_{Vi}(t_{3})\\ \vdots \\ A_{Vi}(t_{T}) \end{array}\right]I_{Hi}+\left[\begin{array}{c} n_{Vi}(t_{2})\\ n_{Vi}(t_{3})\\ \vdots \\ n_{Vi}(t_{T}) \end{array}\right]\\ \to G_{Vi}=\phi_{Vi}R_{Vi}+\varepsilon_{Vi},\;\mathrm{for}\;i=1,2,\cdots ,G^{P} \end{array}$$

Due to the inherent biological mechanism, we need to set constraints on some parameters, such as degradation rate ≤ 0 and translation rate ≥ 0. Thus, real parameters of regulation and interaction order were estimated by solving the following constrained least square problems, respectively:(13)$$\begin{array}{l} {\hat{I}}_{Hi}=\underset{I_{Hi}}{\arg\min}{\left\|\phi_{Hi}I_{Hi}+\varepsilon_{Hi}-P_{Hi}\right\|}_{2}^{2}\\ subject\;to\;\left[\underset{HH_{i}}{\underset{︸}{\left.\begin{array}{ccc} 0 & \cdots & 0\\ 0 & \cdots & 0 \end{array}\right|}}\left.\underset{HP_{i}}{\underset{︸}{\begin{array}{ccc} 0 & \cdots & 0\\ 0 & \cdots & 0 \end{array}}}\right|\begin{array}{ccc} -1 & 0 & 0\\ 0 & 1 & 0 \end{array}\right]{\hat{I}}_{Hi}\leq \left[\begin{array}{c} 0\\ 1 \end{array}\right] \end{array}$$
(14)$$\begin{array}{l} {\hat{I}}_{Pi}=\underset{I_{Hi}}{\arg\min}{\left\|\phi_{Pi}I_{Pi}+\varepsilon_{Pi}-P_{Pi}\right\|}_{2}^{2}\\ subject\;to\;\left[\underset{PH_{i}}{\underset{︸}{\left.\begin{array}{ccc} 0 & \cdots & 0\\ 0 & \cdots & 0 \end{array}\right|}}\left.\underset{PP_{i}}{\underset{︸}{\begin{array}{ccc} 0 & \cdots & 0\\ 0 & \cdots & 0 \end{array}}}\right|\begin{array}{ccc} -1 & 0 & 0\\ 0 & 1 & 0 \end{array}\right]{\hat{I}}_{Pi}\leq \left[\begin{array}{c} 0\\ 1 \end{array}\right] \end{array}$$
(15)$$\begin{array}{l} {\hat{R}}_{Gi}=\underset{R_{Gi}}{\arg\min}{\left\|\phi_{Gi}R_{Gi}+\varepsilon_{Gi}-G_{Hi}\right\|}_{2}^{2}\\ subject\;to\;\left[\underset{GT_{i}}{\underset{︸}{\left.\begin{array}{ccc} 0 & \cdots & 0\\ \vdots & \ddots & \vdots \\ 0 & \cdots & 0\\ 0 & 0 & 0 \end{array}\right|}}\underset{GM_{i}}{\underset{︸}{\left.\begin{array}{ccc} 1 & \cdots & 0\\ \vdots & \ddots & \vdots \\ 0 & \cdots & 1\\ 0 & 0 & 0 \end{array}\right|}}\underset{GL_{i}}{\underset{︸}{\left.\begin{array}{ccc} 0 & \cdots & 0\\ \vdots & \ddots & \vdots \\ 0 & \cdots & 0\\ 0 & 0 & 0 \end{array}\right|}}\begin{array}{cc} 0 & 0\\ \vdots & \vdots \\ 0 & 0\\ 1 & 0 \end{array}\right]{\hat{R}}_{Gi}\leq \left[\begin{array}{c} 0\\ \vdots \\ 0\\ 1 \end{array}\right] \end{array}$$
(16)$$\begin{array}{l} {\hat{R}}_{Vi}=\underset{R_{Vi}}{\arg\min}{\left\|\phi_{Vi}R_{Vi}+\varepsilon_{Vi}-G_{Vi}\right\|}_{2}^{2}\\ subject\;to\;\left[\underset{VT_{i}}{\underset{︸}{\left.\begin{array}{ccc} 0 & \cdots & 0\\ \vdots & \ddots & \vdots \\ 0 & \cdots & 0\\ 0 & 0 & 0 \end{array}\right|}}\underset{VM_{i}}{\underset{︸}{\left.\begin{array}{ccc} 1 & \cdots & 0\\ \vdots & \ddots & \vdots \\ 0 & \cdots & 1\\ 0 & 0 & 0 \end{array}\right|}}\underset{VL_{i}}{\underset{︸}{\left.\begin{array}{ccc} 0 & \cdots & 0\\ \vdots & \ddots & \vdots \\ 0 & \cdots & 0\\ 0 & 0 & 0 \end{array}\right|}}\underset{VV_{i}}{\underset{︸}{\left.\begin{array}{ccc} 0 & \cdots & 0\\ \vdots & \ddots & \vdots \\ 0 & \cdots & 0\\ 0 & 0 & 0 \end{array}\right|}}\begin{array}{cc} 0 & 0\\ \vdots & \vdots \\ 0 & 0\\ 1 & 0 \end{array}\right]{\hat{R}}_{Vi}\leq \left[\begin{array}{c} 0\\ \vdots \\ 0\\ 1 \end{array}\right] \end{array}$$

After solving the above constrained least square problems to estimate parameters of regulations and interactions for each protein, gene, miRNA, and lncRNA of the host cell and hRSV, we purposed the Akaike information criterion (AIC) to prune false positives [8]. AIC considers both the model complexity (order) and estimated error to find the fittest parameter order. The AIC value for each system order was shown as the following equations, and our goal is to find the minimum AIC value as the real parameter order.
(17)$$AIC_{Hi}(HH_{i},HP_{i})=\log(\frac{{\left\|\phi_{Hi}{\hat{I}}_{Hi}+\varepsilon_{Hi}-P_{Hi}\right\|}_{2}^{2}}{T-1})+\frac{2\dim({\hat{I}}_{Hi})}{T-1}$$
(18)$$AIC_{Pi}(PH_{i},PP_{i})=\log(\frac{{\left\|\phi_{Pi}{\hat{I}}_{Pi}+\varepsilon_{Pi}-P_{Pi}\right\|}_{2}^{2}}{T-1})+\frac{2\dim({\hat{I}}_{Pi})}{T-1}$$
(19)$$AIC_{Gi}(GT_{i},GM_{i},GL_{i})=\log(\frac{{\left\|\phi_{Gi}{\hat{R}}_{Gi}+\varepsilon_{Gi}-G_{Hi}\right\|}_{2}^{2}}{T-1})+\frac{2\dim({\hat{R}}_{Gi})}{T-1}$$
(20)$$AIC_{Vi}(VT_{i},VM_{i},VL_{i},VV_{i})=\log(\frac{{\left\|\phi_{Vi}{\hat{R}}_{Vi}+\varepsilon_{Vi}-G_{Vi}\right\|}_{2}^{2}}{T-1})+\frac{2\dim({\hat{R}}_{Vi})}{T-1}$$
where dim(${\hat{I}}_{Hi}$), dim(${\hat{I}}_{Pi}$), dim(${\hat{R}}_{Gi}$), and dim(${\hat{R}}_{Vi}$) denote the parameter vector dimension of each model, respectively. We solved constrained least square Equations (13)–(16) by using the MATLAB lsqlin function and then calculated the AIC value by Equations (17)–(20). Increasing the number of parameters would decrease the least square error term in the AIC equation but it would increase the model complexity (dimension) in the second term. By finding the minimum AIC value [7,8], we would find the trade-off between model complexity and estimated error to find the real order of interactions and regulations of each gene, miRNA and lncRNA of infected cells, and hRSV. The HPI-GWGEN is still very complex and cannot be annotated by KEGG pathways; therefore the principal-network projection (PNP) method is employed to extract the core HPI-GWGEN from real HPI-GWGEN. For the convenience of PNP, the HPI-GWGEN is represented by the following matrix:(21)$$\begin{array}{ll} M & =\left[\begin{array}{llll} I_{HP↔HP} & I_{PP↔HP} & 0_{H\times M} & 0_{H\times L}\\ I_{HP↔PP} & I_{PP↔HP} & 0_{P\times M} & 0_{P\times L}\\ R_{HP\to HG} & 0_{G\times P} & R_{HM\to HG} & R_{HL\to HG}\\ R_{HP\to HM} & 0_{M\times P} & R_{HM\to HM} & R_{HL\to HM}\\ R_{HP\to HL} & 0_{L\times P} & R_{HM\to HL} & R_{HL\to HL}\\ R_{HP\to PG} & R_{PP\to PG}{}^{R} & R_{HM\to PG} & R_{HL\to PG} \end{array}\right]\\ & =\left[\begin{array}{cccc} \begin{array}{ccc} {\hat{I}}_{11}^{HP↔HP} & \cdots & {\hat{I}}_{H1}^{HP↔HP}\\ \vdots & \ddots & \vdots \\ {\hat{I}}_{1H}^{HP↔HP} & \cdots & {\hat{I}}_{HH}^{HP↔HP} \end{array} & \begin{array}{ccc} {\hat{I}}_{11}^{PP↔HP} & \cdots & {\hat{I}}_{P1}^{PP↔HP}\\ \vdots & \ddots & \vdots \\ {\hat{I}}_{1H}^{PP↔HP} & \cdots & {\hat{I}}_{PH}^{PP↔HP} \end{array} & 0_{H\times M} & 0_{H\times L}\\ \begin{array}{ccc} {\hat{I}}_{11}^{HP↔PP} & \cdots & {\hat{I}}_{H1}^{HP↔PP}\\ \vdots & \ddots & \vdots \\ {\hat{I}}_{1P}^{HP↔PP} & \cdots & {\hat{I}}_{HP}^{HP↔PP} \end{array} & \begin{array}{ccc} {\hat{I}}_{11}^{PP↔HP} & \cdots & {\hat{I}}_{P1}^{PP↔HP}\\ \vdots & \ddots & \vdots \\ {\hat{I}}_{1P}^{PP↔HP} & \cdots & {\hat{I}}_{PP}^{PP↔HP} \end{array} & 0_{P\times M} & 0_{P\times L}\\ \begin{array}{ccc} {\hat{I}}_{11}^{HP\to HG} & \cdots & {\hat{I}}_{H1}^{HP\to HG}\\ \vdots & \ddots & \vdots \\ {\hat{I}}_{1G}^{HP\to HG} & \cdots & {\hat{I}}_{HG}^{HP\to HG} \end{array} & 0_{G\times P} & \begin{array}{ccc} {\hat{I}}_{11}^{HM\to HG} & \cdots & {\hat{I}}_{M1}^{HM\to HG}\\ \vdots & \ddots & \vdots \\ {\hat{I}}_{1G}^{HM\to HG} & \cdots & {\hat{I}}_{MG}^{HM\to HG} \end{array} & \begin{array}{ccc} {\hat{I}}_{11}^{HL\to HG} & \cdots & {\hat{I}}_{L1}^{HL\to HG}\\ \vdots & \ddots & \vdots \\ {\hat{I}}_{1G}^{HL\to HG} & \cdots & {\hat{I}}_{LG}^{HL\to HG} \end{array}\\ \begin{array}{ccc} {\hat{I}}_{11}^{HP\to HM} & \cdots & {\hat{I}}_{H1}^{HP\to HM}\\ \vdots & \ddots & \vdots \\ {\hat{I}}_{1M}^{HP\to HM} & \cdots & {\hat{I}}_{HM}^{HP\to HM} \end{array} & 0_{M\times P} & \begin{array}{ccc} {\hat{I}}_{11}^{HM\to HM} & \cdots & {\hat{I}}_{M1}^{HM\to HM}\\ \vdots & \ddots & \vdots \\ {\hat{I}}_{1M}^{HM\to HM} & \cdots & {\hat{I}}_{MM}^{HM\to HM} \end{array} & \begin{array}{ccc} {\hat{I}}_{11}^{HL\to HM} & \cdots & {\hat{I}}_{L1}^{HL\to HM}\\ \vdots & \ddots & \vdots \\ {\hat{I}}_{1M}^{HL\to HM} & \cdots & {\hat{I}}_{LM}^{HL\to HM} \end{array}\\ \begin{array}{ccc} {\hat{I}}_{11}^{HP\to HL} & \cdots & {\hat{I}}_{H1}^{HP\to HL}\\ \vdots & \ddots & \vdots \\ {\hat{I}}_{1L}^{HP\to HL} & \cdots & {\hat{I}}_{HL}^{HP\to HL} \end{array} & 0_{L\times P} & \begin{array}{ccc} {\hat{I}}_{11}^{HM\to HL} & \cdots & {\hat{I}}_{M1}^{HM\to HL}\\ \vdots & \ddots & \vdots \\ {\hat{I}}_{1L}^{HM\to HL} & \cdots & {\hat{I}}_{ML}^{HM\to HL} \end{array} & \begin{array}{ccc} {\hat{I}}_{11}^{HL\to HL} & \cdots & {\hat{I}}_{L1}^{HL\to HL}\\ \vdots & \ddots & \vdots \\ {\hat{I}}_{1L}^{HL\to HL} & \cdots & {\hat{I}}_{LL}^{HL\to HL} \end{array}\\ \begin{array}{ccc} {\hat{I}}_{11}^{HP\to PG} & \cdots & {\hat{I}}_{H1}^{HP\to PG}\\ \vdots & \ddots & \vdots \\ {\hat{I}}_{1V}^{HP\to PG} & \cdots & {\hat{I}}_{HV}^{HP\to PG} \end{array} & \begin{array}{ccc} {\hat{I}}_{11}^{PP\to PG} & \cdots & {\hat{I}}_{P1}^{PP\to PG}\\ \vdots & \ddots & \vdots \\ {\hat{I}}_{1V}^{PP\to PG} & \cdots & {\hat{I}}_{PV}^{PP\to PG} \end{array} & \begin{array}{ccc} {\hat{I}}_{11}^{HM\to PG} & \cdots & {\hat{I}}_{M1}^{HM\to PG}\\ \vdots & \ddots & \vdots \\ {\hat{I}}_{1V}^{HM\to PG} & \cdots & {\hat{I}}_{MV}^{HM\to PG} \end{array} & \begin{array}{ccc} {\hat{I}}_{11}^{HL\to PG} & \cdots & {\hat{I}}_{L1}^{HL\to PG}\\ \vdots & \ddots & \vdots \\ {\hat{I}}_{1V}^{HL\to PG} & \cdots & {\hat{I}}_{LV}^{HL\to PG} \end{array} \end{array}\right] \end{array}$$

In Equation (21), *HP*, *PP*, *HG*, *HL*, *HM,* and *PG* represent host protein, pathogen protein, host gene, host lncRNA, host miRNA, and pathogen gene, respectively; *I* and *R* represent interaction ability in PPIN and regulation ability in GRN, respectively. *H*, *P*, *G*, *M*, *L*, and *V* represent the number of host protein, pathogen protein, host gene, host lncRNA, host miRNA, and pathogen gene, respectively.

### 2.6. Extracting Core HPI-GWGEN via Principal-Network Projection

Since real HPI-GWGEN was still too complex to be annotated by KEGG pathways to investigate the pathogenic mechanism of hRSV infection, we purposed principal-network projection methods to extract the core HPI-GWGEN. The PNP method is an application of singular-value decomposition [21] to rank each node (genes, proteins, miRNA, and lncRNA) in HPI-GWGEN so as to know the significance of each node in HPI-GWGEN. First, we did the singular value decomposition (SVD) on HPI-GWGEN as follows:(22)M=USVTU∈ℝ(H+P+G+M+L+V×H+P+G+M+L+V)S∈ℝ(H+P+G+M+L+V×H+P+M+L)V∈ℝ(H+P+M+L×H+P+M+L)
where *H*, *P*, *G*, *M*, *L*, and *V* represent the number of host protein, pathogen protein, host gene, host miRNA, host lncRNA, and pathogen (virus) gene, respectively.

In order to extract the core HPI-GWGEN, we chose top-*k* singular values in *S* which contain 85% energy, and top-*k* singular vectors of *U* and *V* which consist of the principal-network structure of HPI-GWGEN matrix *M* as core HPI-GWGEN. The top-*k* singular values satisfy with the following inequality:(23)$$\frac{\sum_{j=1}^{k}s_{jj}^{2}}{\sum_{i=1}^{H+P+M+L}s_{ii}^{2}}\geq 0.85$$

Then, in order to rank each node (row) of HPI-GWGEN *M* in (21) and obtain the projection value on the core HPI-GWGEN, we define the projection value for each row $M_{i}$ as follows:(24)$$proj_{i}=\sqrt{{\sum_{j=1}^{k}\left(M_{i}V_{J}^{T}\right)}^{2}},for\;i=1,\ldots ,R$$
where $M_{i}$ and $V_{j}{}^{T}$ represent the ith row of matrix *M* and the jth column of singular matrix *V*, respectively. *R* represents the row number of matrix *M* which means the total number of all nodes in HPI-GWGEN (i.e., *H* + *P* + *G* + *M* + *L* + *V*).

At the end, we finally obtained the projection value of each node, and we could rank them based on their significance of each node in HPI-GWGEN to obtain the top-6000 significant nodes to consist of core HPI-GWGEN. Then, we proposed DAVID to help us do KEGG enrichment analysis by top-6000 significant genes, proteins, miRNA, and lncRNA in core HPI-GWGEN. With the aid of KEGG enrichment analysis and annotation of the core HPI-GWGEN to obtain core HPI signaling pathways in Figure 4 by KEGG pathways, we investigated the pathogenic mechanism for hRSV infection. By the core HPI signaling pathways, we selected five significant biomarkers as our drug target for hRSV infection treatment.

### 2.7. Systematic Drug Repurposing Design of hRSV Infection via DNN-Based DTI Model and Drug Specifications

After obtaining the five significant biomarkers as drug targets for hRSV infection treatment, the DNN-DTI model is employed to be trained by drug–target databases to predict the potential molecular drugs for these five significant biomarkers of hRSV infection. Based on the prediction of the DNN-based DTI model, we chose several candidate drugs which can target these five drug targets for hRSV infection treatment. Due to the complexity of drug mechanisms in human beings, we purposed three drug design specifications (i.e., regulation ability, sensitivity, and toxicity) to make our candidate drug helpful in hRSV infection treatment. The flowchart of the drug-discovery design is shown in Figure 2.

#### 2.7.1. DNN-Based DTI Model for Drug Repurposing of hRSV Infection

Before training the DNN model as a DTI model by drug–target interaction database to predict potential molecular drugs for each biomarker of hRSV infection, we first preprocessed the drug–target interaction (DTI) data as shown in Figure 2. We collected the drug–target interaction data from several DTI databases including ChEMBL, BindingDB, Pubchem, UniProt, and DrugBank. In order to input these data into the DNN model, we must transform them into numerical vectors. We utilized PyBioMed to transform drug chemical structures and protein sequences into numerical feature vectors. At the same time, the drug feature vector and protein feature vector were concatenated as the following drug–target vector:(25)$$F_{DT}=[F_{D},F_{T}]$$
where $F_{D}$ and $F_{T}$ represent the drug feature vector and target (gene) feature vector. The drug–target feature vector $F_{DT}$ would be the input of the DNN-based DTI model.

The collected DTI data includes 80,291 positive interactions and 100,294 unknown as negative interactions. Due to the imbalance of the dataset, we randomly downsampled the negative interaction dataset to 80,291. Then, we divided the whole dataset into a training set (fourth three) and a testing set (fourth one). As all the feature vectors were located in different scales, we applied standardization to each training data to solve this problem. Moreover, because of the complexity of each drug–target feature vector, it might let the DNN model hard-learn the features or increase the computational complexity of the DNN model. We proposed PCA to reduce the dimension of every feature vector. Every drug–target feature vector was reduced to 1000 dimensions as the input layer of 1000 DNN neurons, as shown in Figure 2. 

For the deep neural-network model, we employed four hidden layers which contained 512, 256, 128, and 64 neurons, respectively. In a hidden layer, we used ReLU as the activation function and dropout set 0.2 to avoid overfitting [22]. The ReLU function has a strong biological underpinning [23] and helps the DNN model learn the nonlinearity. As drug–target interaction basically was a binary classification problem, the output layer only contained one neuron to indicate the probability of drug–target interaction.

As we discussed before, DTI was a binary classification problem. We proposed binary cross-entropy as the loss function which was shown as the following equation:(26)$$L(p,\hat{p})=-\frac{1}{N}\sum_{n=1}^{N}\left(p_{n}\times \log({\hat{p}}_{n})+(1-p_{n})\times \log(1-{\hat{p}}_{n})\right)$$
where $p_{n}$ and ${\hat{p}}_{n}$ represent whether the *n*th sample is interacted and the probability of the nth sample predicted in the DNN model, respectively. After defining the loss function, we applied ADAM [24] as our backward propagation algorithm and completed the whole DNN-DTI model architecture. The model is trained by Keras with 64 batch size and 200 epochs, and we also proposed 5-fold validation to evaluate the predicted performance of the trained model. To visualize the high performance of the DNN-based DTI model, we plotted the receiver operating characteristic (ROC) curve in Figure 6. ROC curve is used for the binary classification model and is aimed to examine whether the model can distinguish the positive and negative sample. The area under the ROC curve (AUC), which was shown in Figure 6, was also applied for a benchmark in the binary classification problem, and the higher the AUC value, the model performance is better.

#### 2.7.2. Drug Design Specifications

After obtaining the candidate molecular drugs for five biomarkers by the prediction of the proposed DNN-based DTI model, we started considering the reliability of these drugs, so we purposed three benchmarks as design specifications, including regulation ability, toxicity, and sensitivity to make sure the drugs were reliable. The LINCS L1000 database [25] is used for the specification of regulation ability and the PRISM database [26] is used for the specification of sensitivity. The sensitivity indicates the drug utility perturbation for human cells. The most important of all is toxicity, and we employed ADMETlab 2.0 [27] to specify the toxicity (LC50). LC50 is the abbreviation of lethal concentration 50%, which means the higher the LC50 value is, the lower toxicity for the human being.

## 3. Result

### 3.1. Extracting Core Signaling Pathways via System Identification and Principal-Network Projection Approach

The overall flowchart is shown in Figure 1. First, we constructed candidate HPI-GWGEN by database mining and integration. Then, we proposed a system identification approach to eliminate the false positives in candidate HPI-GWGEN and then obtained the real HPI-GWGEN. The node and edge number between the candidate and real HPI-GWGEN are shown in Table 1. After the deletion of the false positives in candidate HPI-GWGEN for real HPI-GWGEN of hRSV infection, we still need to apply the PNP method because the real HPI-GWGEN is still pretty complex for the annotation of KEGG pathways. Therefore, we used the PNP method to extract core HPI-GWGEN with 85% energy of real HPI-GWGEN and rank all nodes (genes and proteins) according to their projection (significance) values. To visualize the real HPI-GWGEN and core HPI-GWGEN, we used the software Cytoscape [28] to visualize the whole networks and help us intuitively understand the power of PNP, as shown in Figure 3. We also uploaded top-6000-significance nodes to the DAVID functional annotation tool [29] to help us investigate core HPI signaling pathways for the pathogenic mechanism of the hRSV infecting progression. DAVID presented the enrichment analysis of KEGG which indicates that hRSV infection may involve in what kinds of pathways. The KEGG enrichment analysis is shown in Table 2.

### 3.2. Investigation of Core HPI Signaling Pathways for Pathogenic Mechanism of hRSV Infection Progression

According to the KEGG enrichment analysis of core HPI-GWGEN and the annotation of KEGG pathways, we found the core HPI signaling pathways in the hRSV infection progression, as shown in Figure 4. With the aid of core HPI signaling pathways and their downstream target genes, we could investigate the pathogenic mechanism of hRSV infection and find key biomarkers to help us design a multimolecule drug for therapeutic treatment of hRSV infection in the next step. In this section, we introduced these core HPI signaling pathways and the potential pathogenic mechanism of the biomarkers of hRSV infection.

#### 3.2.1. The Significant Signaling Pathways Involved with Biomarkers TRAF6 and RELA

As shown in Figure 4, TRAF6 is activated by both the IL-17α pathway and the TLR2 pathway. IL-17α is stimulated by the microenvironment and interacted with its receptor IL-17Rα. The second pathway involved in TRAF6 is the TLR2/MyD88 pathway. RSV utilizes receptor TLR2 for launching a proinflammatory response in the RSV infected progression [30]. RSV G protein binds to receptor TLR2 and then activates with MyD88 [31]. MyD88 is an adaptor protein that mediates toll and interleukin (IL)-1 receptor signaling [32]. TRAF6 interacts with signaling proteins MAP2K1, MAPK8, CEBPB and the MAP3K7–TAB2 complex. For its downstream, TRAF6 finally activated TFs FOXO3, RELA, JUN, and CEBPB, respectively. TF FOXO3 is one of the FOXO family members that promotes the expression of cyclin-dependent kinase-inhibitor genes *CDKN1A* and *IL6*. Target gene *CDKN1A* triggers a proliferation arrest [33]. In addition to TFs, FOXO3 and RELA upregulated *IL6*; the mircoRNA LET-7i also directly inhibits IL6 expression. Ligand IL6 induces phosphorylation and nuclear entry of the STAT3 [34] and then IL6 causes a positive feedback loop during hRSV infection [35]. In the IL6 feedback loop, there is also an important TF called RELA. RELA/NF-κB is key in innate inflammation, controlling the expression of inflammatory chemokines as well as mucosal interferons (IFNs) through a process of regulated transcriptional elongation [36]. TF RELA upregulates the target genes *ICAM1*, *AP-1*, *IL6*, and *MMP9.* Although we do not identify any virus gene binding to *ICAM1*, one study shows that *ICAM1* facilitates RSV entry and infection of human epithelial cells by binding to its F protein which means that *ICAM1* is significant for viral replication [37]. According to the severity of hRSV determined by environmental exposure to lipopolysaccharide (LPS), we also find LPS induces *ICAM1*, which is a positive correlation. Nuclear transcription factor JUN was involved in the activation of many cellular functions, including cell proliferation and apoptosis [38]. In clinical trials, *MMP9* demonstrates that it is involved in the antiviral and anti-inflammatory effects [39].

In summary, ligand IL17α and hRSV protein G activate these two signaling pathways, i.e., IL-17α pathway and TLR2 pathway; then, these signals are transducted through their downstream signaling protein TRAF6. TRAF6 interacted with many downstream target genes involved in the upregulations of inflammation, cell proliferation, host defense, and the downregulations of apoptosis. Especially *IL6* forms a positive feedback loop to influence on IL6 pathways and its downstream signaling proteins, STAT3 and TF RELA, which also involve in another significant pathway for hRSV infection, i.e., the RIG-1/MAVS pathway [40]. As TRAF6 and RELA are located at the pivot of two signaling pathways and are involved in many apoptosis-related, LPS-related, proliferation-related target genes in HPI signaling pathways, we considered TRAF6 and RELA as biomarkers of hRSV infection. The RIG-1/MAVS pathway will be discussed in the following section in detail.

#### 3.2.2. The Significant Signaling Pathways Involved with Biomarkers MAVS and IRF3

In this section, we will discuss the RIG-1/MAVS pathway. RIG-1/MAVS pathway is the initial intracellular sensor for hRSV infection and the upstream of the canonical NF-κB pathway [41]. RIG-I is upstream of the MAVS protein which is known as an interferon-β promoter stimulator or virus-induced signaling adaptor [42]. RIG-1 also senses nucleic acids derived from viruses and triggers antiviral innate immune responses [43]. It means RIG-1 plays a vital role in hRSV infection. According to the core HPI signaling pathways, as shown in Figure 4, hRSV NS1 protein was identified as interacting with MAVS which was consistent with a recent study [44]. The interaction of NS1 and MAVS disrupts the RIG-1/MAVS pathway and the downstream TF IFNA1 and target gene IL6, which leads to the upregulations of inflammation [44]. We also proved that TRAF3 interacts with hRSV NS2 protein in a recent study [45] by the proposed system-identification approach. MAVS interacted with TRAF3, and TRAF3 interacted with the TKB1–IKB1CE complex which resulted in the phosphorylation of IRF3. Further both the TKB1–IKB1CE complex and ISG15 interacted with IRF3 too. Therefore, IRF3 was viewed as another biomarker in our signaling pathways because of the multi-interaction in this pathway. IRF3 upregulates the target gene CXCL8, TF JUN, and IFNA1. The identification of high expression of CXCL8 is consistent with a recent study [46]. Target gene CXCL8 is a key driver of the antiviral inflammatory response during hRSV infection [46] and CXCL8 also correlates positively with disease severity during RSV infection [47]. The phosphorylation of IRF3 leads to its dimerization and translocation to the nucleus, where it drives the expression of IFNA1. After upregulating IFNA1, IFNA1 also affects another pathway and another biomarker TYK2. We will discuss the IFNA1 pathway in detail in the following section.

#### 3.2.3. The Significant Signaling Pathways Involved with Biomarker TYK2

Tyrosine kinase 2 (TYK2), a kinase belonging to the JAK family, is constitutively bound to receptor IFNAR1 [48]. From the upstream of the IFN pathway, ligand IFNA1 interacts with receptor IFNAR1 because of the RIG-1/MAVS pathway or the microenvironment. Subsequently, TYK2 phosphorylates STAT2 and STAT1 and binds to TF IRF9 [49]. TF IRF9 translocates to the nucleus and upregulates target genes *OAS2* to upregulate host defense and *CDKN1A* to downregulate the apoptosis. A recent study showed that the inhibition of *OAS2* expression can inhibit the antiviral effect of IFN against hRSV [50]. *CDKN1A* is a target gene involved in proliferation arrest [51]. TYK2 also interacts with STAT3, which is vital to virus infection. TYK2 induces STAT3 phosphorylation and then STAT3 interacts with RELA, which is another significant biomarker [52]. According to a recent study, patients with STAT3 mutation presented with more viral infections [53]. Based on the above discussion, we can explain why the IFN signaling pathway is so important in hRSV infection and why TYK2 is selected as a biomarker in the pathogenic mechanism of hRSV infection.

#### 3.2.4. TNF Signaling Pathway

Tumor necrosis factor TNF is a proinflammatory cytokine that plays a vital role in the innate host defense [54]. As we know, RELA activation in infected cells is important to ligand TNF production [55]. Thus, the activation of RELA leads to the activation of the TNF signaling pathway. At the downstream of the TNF signaling pathway, TFs CREB1, CEBPB, and JUN were activated. CREB1 can upregulate with target genes *ICAM1*, *FOS*, *IL6,* and *MMP9*. CEBPB can upregulate with target genes *ICAM1*, *FOS,* and *CXCL8*. A recent study shows that a replicating virus and RELA activation are required for the high expression of CXCL8 [56]. CXCL8 is mainly involved in the initiation and amplification of acute inflammatory reactions [57]. The high expression of IL6 also inhibits cell apoptosis which has an advantage to viral replicating. From the above analysis, we can know that RELA activation induces the TNF signaling pathway, and then the TNF signaling pathway can upregulate the target genes *ICAM1*, *FOS*, *IL6,* and *CXCL8* to cause the upregulation of inflammation and the downregulation of apoptosis. IL6 is a cytokine that transmits defense signals from a pathogen invasion or tissue damage site to stimulate acute phase reactions and immune responses to prepare for host defense [58]. Since the TNF signaling pathway targets *IL6*, and *IL6* transmits a signal to RELA, the activation of RELA, the production of TNF, and the high expression of IL6 form a positive cyclic signaling pathway.

#### 3.2.5. Conclusion of HPI Signaling Pathways during hRSV Infection

First, we discussed IL17 and TLR2 pathways. The hRSV G protein first interacted with TLR2 and then upregulated IL6 at the downstream. After the activation of the IL6 pathway, ligand IL6 interacted with STAT3 indirectly and then interacted with TF RELA. RELA also induces the production of TNF, and then the TNF signaling pathway targets *IL6* at the downstream. This process induced a positive feedback loop during the hRSV infection. Subsequently, we concluded the RIG-1/MAVS pathway was important to viral replication. In this signaling pathway, hRSV protein NS1 interacted with MAVS, and hRSV protein NS2 interacted with IRF3. At the downstream of the pathway, IRF3 upregulated TF IFNA1, and the IFN pathway was activated too. The IFN pathway upregulated target genes *OAS2* and *CDKN1A* which led to the regulation of cellular functions, i.e., apoptosis, cell proliferation, and virus defense. Based on the above discussion, we could know that all these signaling pathways were highly correlated via analyzing the core HPI signaling pathway during hRSV infection.

### 3.3. Multimolecule Drug Repurposing by DNN-Based DTI Model and Drug Design Specifications

After identifying significant biomarkers TYK2, RELA, IRF3, TRAF6, and MAVS as drug targets for the therapeutic treatment of hRSV infection, we purposed the DNN-based DTI model to discover potential multimolecule drugs for our drug targets. The proposed DNN-based DTI model was good at predicting unknown drug-target pairs for these drug targets (biomarkers) after the training via DTI databases, as discussed in Section 2.7 and shown in Figure 2. After candidate molecular drugs were predicted for these five significant biomarkers by the well-trained DNN-based DTI model in Section 2.7, we filtered these potential drugs from the candidate molecular drugs with three drug design specifications including regulation ability, sensitivity, and toxicity as a multimolecule drug for a further clinical trial.

#### 3.3.1. DNN-Based DTI Model

The flowchart of multimolecular drug design via the DNN-based DTI model trained by DTI databases as discussed in Section 2.7 is shown in Figure 2. The architecture of DNN consisted of one input layer, four hidden layers, and one output layer. We set batch size as 64, epoch as 200, and Adam algorithm [24] as optimizer. We also used fivefold cross-validation to evaluate the performance of the DNN-based DTI model to prevent overfitting, which is shown in Table 3. The accuracy and loss followed by each epoch are shown in Figure 5. Since the DNN-based DTI model is binary classification essentially, we used the ROC curve and AUC to evaluate the drug-target prediction performance by the DNN-based DTI model. The ROC curve is shown in Figure 6 and AUC is 0.981, which indicates that the drug-target prediction performance of the proposed DNN-based DTI model is powerful in the drug-target binary classification problem.

#### 3.3.2. Multimolecule Drug Repurposing for hRSV Infection Treatment

After training the DNN-DTI model by DTI data in the DTI database, as shown in Figure 2, we can predict candidate molecular drugs for five biomarkers as drug targets. Due to the complexity of the drug mechanism, we screened candidate molecular drugs with three drug specifications, including regulation ability, sensitivity, and toxicity, simultaneously, as shown in Table 4. The multimolecule drug we selected, based on three design specifications and their interactions with biomarkers, was shown in Table 5.

## 4. Discussion

Although hRSV infection affects tens of millions of people each year, its pathogenic mechanism is barely known. Nowadays, there is still no vaccine or effective medicine to prevent hRSV infection. In this study, we first constructed a candidate HPI-GWGEN by big-data mining and used the system-identification method by two-side host/pathogen RNA-seq data to identify the true system-parameter orders to prune false positives in candidate HPI-GWGEN as the real HPI-GWGEN in the hRSV infection. Secondly, we used the PNP method to extract the significant part of the network matrix for core HPI-GWGEN to further annotate as core HPI signaling pathways by KEGG pathways. In the third step, the core HPI signaling pathways were employed to investigate the pathogenic mechanism during hRSV infection and identify significant biomarkers for therapeutic treatment. The fourth step was to predict candidate molecule drugs for these significant biomarkers via training the DNN-based DTI model by DTI data in DTI databases. The final step aimed to select potential molecule drugs by three drug design specifications for these significant biomarkers to combine as a multimolecule drug for the therapeutic treatment of hRSV infection.

At the end, we selected three kinds of molecular drugs to combine as a multimolecule drug for hRSV infection treatment, which contained acitretin, RS-67333, and phenformin. Acitretin is a retinoic acid derivative, and it has been approved by the US Food and Drug Administration (FDA) [59]. Traditionally, acitretin is an effective treatment for psoriasis [60] and can enhance the RIG-1 signaling pathway [59]. A recent study even showed that STAT3 and STAT1 were downregulated by acitretin [61] and these studies all indicated a high correlation with HPI signaling pathways for hRSV infection. RS-67333 has been investigated as a potential rapid-acting antidepressant, nootropic, and treatment for Alzheimer’s disease. RS-67333 can reduce the expression levels of *IL6* and TNF [62]. As we have discussed in the previous section, IL6 could cause a positive-feedback loop, which means that downregulating IL6 is pretty important for the hRSV infection treatment. Phenformin was introduced in the U.S. in 1957 to treat noninsulin-dependent diabetes mellitus (NIDDM). Although phenformin was withdrawn in 1977 because of the high incidence of associated lactic acidosis [63], more and more studies repurposed phenformin as the breast cancer and COVID-19 treatment [64,65]. Given the shorter duration of hRSV infection treatment, if the treatment time and dose are shortened and the patient’s renal function is pre-examined [66], phenformin may be a good drug for the hRSV infection treatment.

Finally, we discussed the drug–drug interaction of multimolecule drugs. Although drug–drug interactions constitute only a small proportion of adverse drug reactions, they are still important [67]. We used Medscape, WebMD, and Drug.com [68] to find the interactions between three drugs and there were no interactions in their database, which indicates the availability of selected molecular drugs as multimolecule drugs for hRSV infection treatment.

## 5. Conclusions

We purposed a systems biology approach by host/pathogen RNA-Seq data to construct core signaling pathways to investigate the pathogenic mechanism of hRSV infection. Then, based on the pathogenic mechanism of hRSV infection, we selected five biomarkers, including TRAF6, IRF3, MAVS, IL6, and STAT3, as our drug targets. By using the systematic drug-discovery method, based on potential molecular drugs of these five biomarkers via the prediction of a well-trained DNN-based DTI model by DTI data in the DTI database, we selected three multimolecule drugs, including acitretin, RS-67333, and phenformin, based on three drug design specifications, including regulation ability, sensitivity, and toxicity as the multimolecule drug of hRSV infection. To the best of our knowledge, these three molecular drugs have no drug–drug interaction. These three molecular drugs can also regulate all biomarkers in hRSV infection.

## Figures and Tables

**Figure 1 biomedicines-11-01531-f001:**
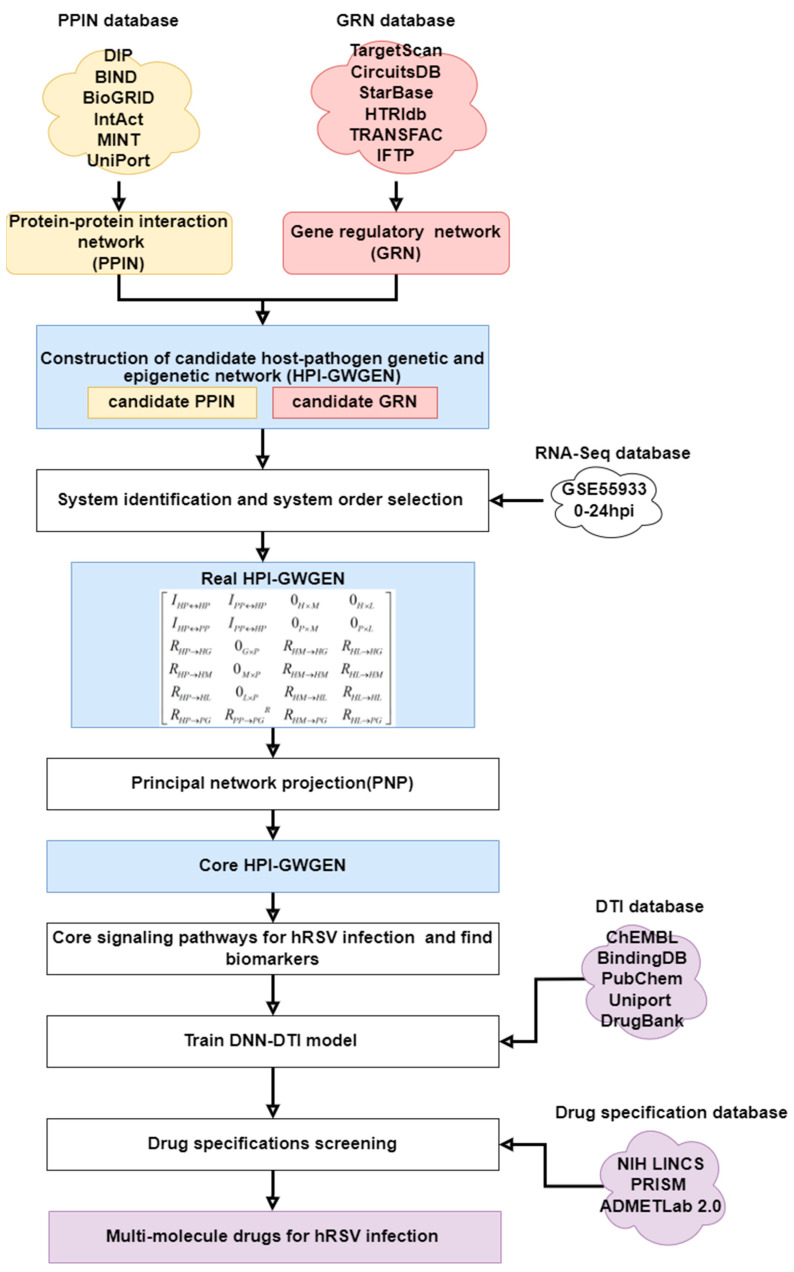
The whole flowchart of the systematic drug discovery method by big-data mining and dynamic modeling of PPIN and GRN for investigating hRSV pathogenic mechanism by two-side RNA-Seq data of hRSV and human A549 cell via systems biology methods and deep-learning approach, including system identification, DNN-based DTI model, and drug-specification screening to select molecular drugs which can target significant biomarkers can comprise a multimolecule drug as the clinical-trial recommendation for hRSV treatment.

**Figure 2 biomedicines-11-01531-f002:**
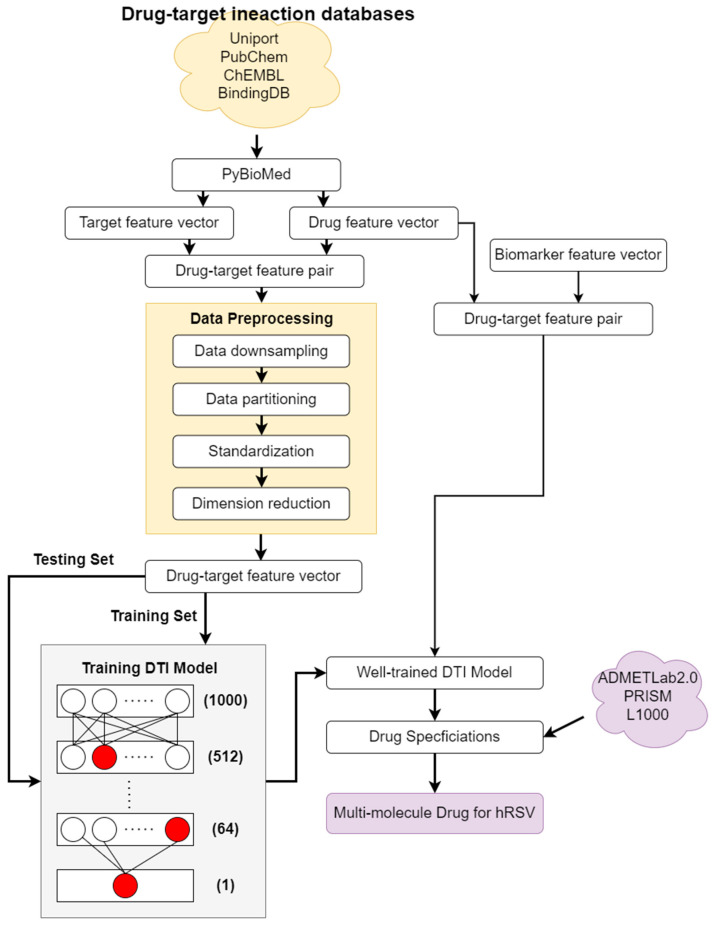
The flowchart of multimolecule drug design for hRSV infection treatment. At the beginning, we integrated drug–target databases, transformed them to feature vectors, and performed data preprocessing. Then, we employed drug–target pairs in DTI databases to train DNN as a DTI model. At the end, we used the well-trained DTI model to predict the candidate molecular drugs for each drug target (biomarker) in Table 4 and then filtered the candidate molecular drugs with three drug specifications as potential molecular drugs. These potential molecular drugs are selected to consist of a multimolecular drug to target these biomarkers as shown in Table 5.

**Figure 3 biomedicines-11-01531-f003:**
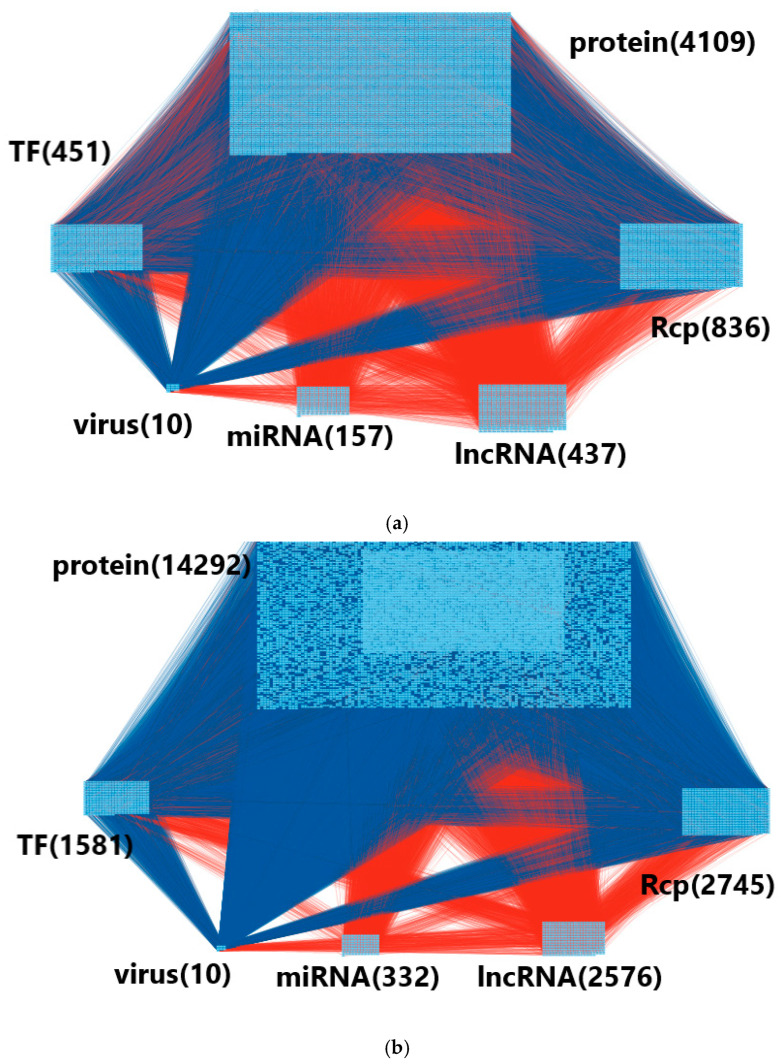
(**a**) Visualization of core HPI-GWGEN of hRSV infection. (**b**) Visualization of real HPI-GWGEN of hRSV infection. The real HPI-GWGEN of hRSV infection is obtained by pruning false positives from candidate HPI-GWGEN by AIC system order detection and system identification method via two-side HPI time-profile RNA-Seq data, and then core HPI-GWGEN is obtained by 6000 significant nodes of real HPI-GWGEN by PNP method in (22)–(24). The blue lines represent the protein–protein interaction, and the red lines represent the gene-regulation relationship. The number of proteins, Rcp (receptor), miRNA, lncRNA, virus gene, and transcription factor (TF) are shown in the figure.

**Figure 4 biomedicines-11-01531-f004:**
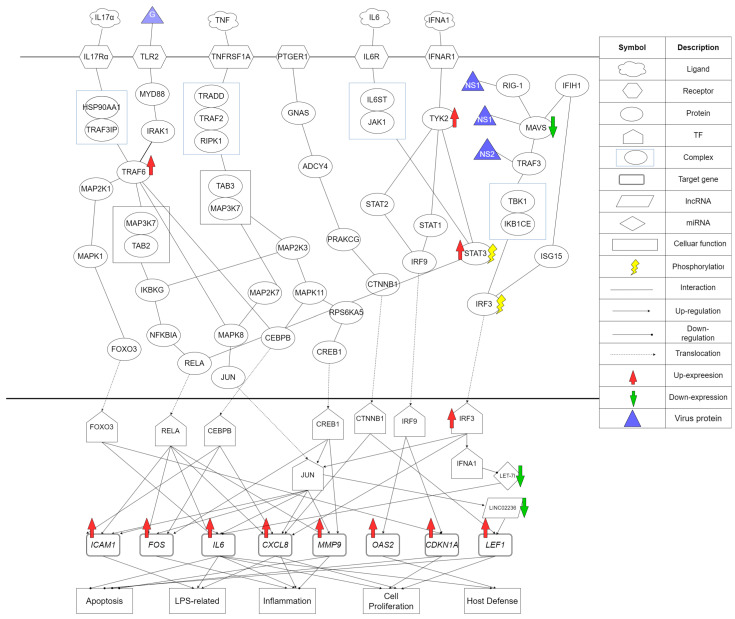
The core HPI signaling pathways of core HPI-GWGEN in Figure 3a, based on the annotation of the KEGG pathways during hRSV infection. These core HPI signaling pathways lead to the upregulations of downstream target genes and consequently regulation of the apoptosis, LPS related, inflammation, cell proliferation, and host defense. The table on the right side describes the meaning of each symbol.

**Figure 5 biomedicines-11-01531-f005:**
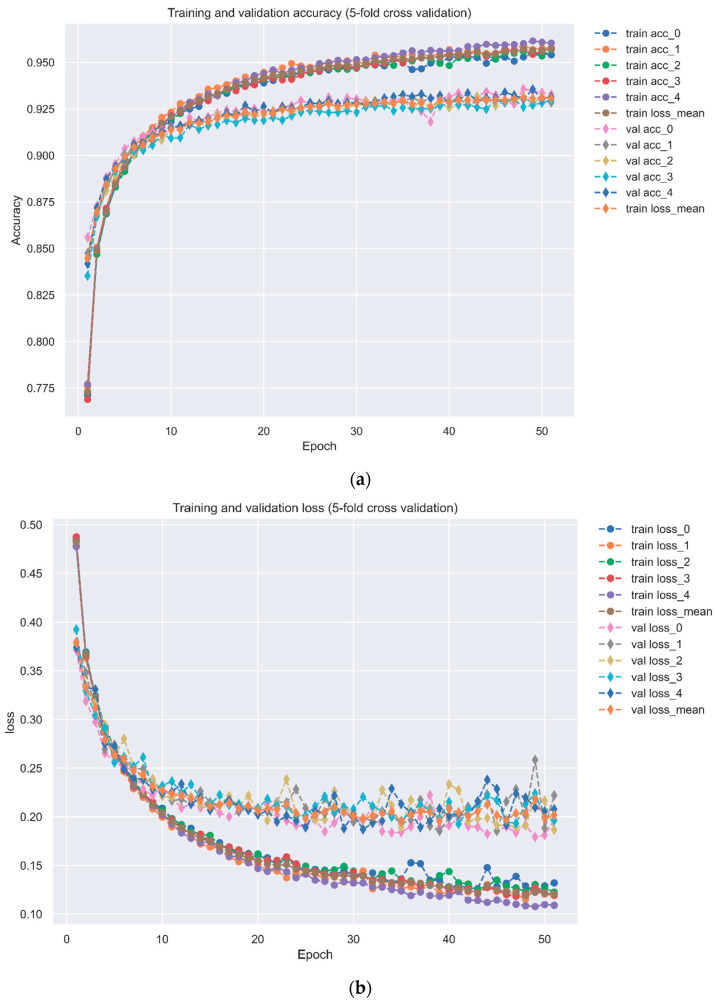
(**a**) The accuracy performance of DNN-based DTI model by 5-fold cross-validation. Early stopping is applied at epoch 51 to avoid overfitting. (**b**) The loss performance of drug-target prediction through DNN-based DTI model by 5-fold cross-validation. Early stopping is applied at epoch 51 to avoid overfitting.

**Figure 6 biomedicines-11-01531-f006:**
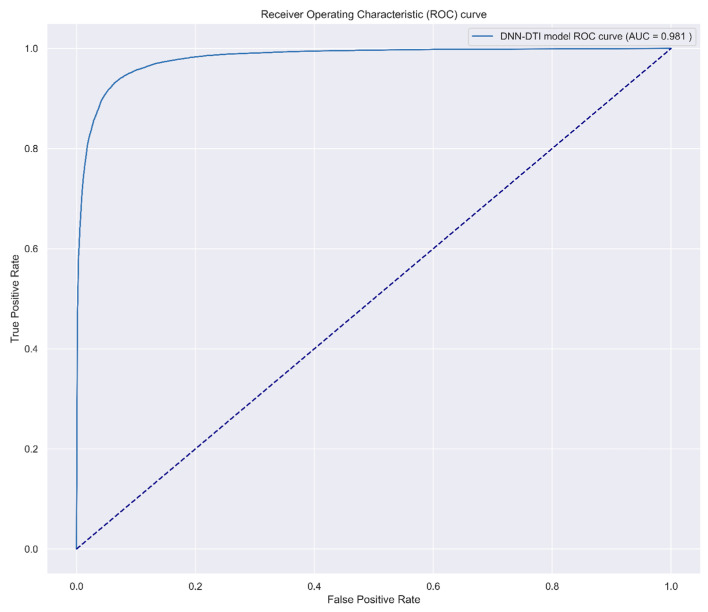
The receiver operating characteristic (ROC) curve for the drug-target prediction performance of the DNN-based DTI model. True positive rate (y axis) is also known as sensitivity, which refers to the probability of accuracy in the positive dataset. False-positive rate (x axis) refers to the probability of failure in the negative dataset. Area under curve (AUC) value indicates the performance of separability. The higher AUC value means the model has better separability, i.e., the better drug-target prediction for drug targets (biomarkers).

**Table 1 biomedicines-11-01531-t001:** The number of nodes and edges in candidate and real HPI-GWGEN of hRSV infection. The real HPI-GWGEN is obtained by pruning false positives in candidate HPI-GWGEN through AIC as real parameter order in (17)–(20) via the system-identification method by two-side HPI RNA-Seq data.

**Nodes**	**Candidate GWGEN**	**Real GWGEN**
Proteins	14,389	14,292
Receptors	3082	2745
Transcription factors	1594	1581
miRNAs	551	332
LncRNAs	3245	2576
Virus	10	10
Total nodes	22,871	21,536
**Edges**	**Candidate GWGEN**	**Real GWGEN**
PPIs	4,272,492	748,964
TF–Receptor	1459	868
TF–TF	841	557
TF–Protein	5326	3635
TF–miRNA	264	138
TF–lncRNA	1338	827
TF–Virus	15,940	76
miRNA–Receptor	140	26
miRNA–TF	62	15
miRNA–Protein	462	134
miRNA–miRNA	18	4
miRNA–lncRNA	137	33
miRNA–Virus	5510	24
lncRNA–Receptor	4070	2020
lncRNA–TF	2126	1213
lncRNA–Protein	14,646	8598
lncRNA–miRNA	771	336
lncRNA–lncRNA	3751	2118
lncRNA–Virus	32,450	221
Virus–Virus	90	2
Total edges	4,361,893	769,809

**Table 2 biomedicines-11-01531-t002:** The KEGG enrichment analysis for core HPI-GWGEN of hRSV infection by DAVID.

KEGG Pathway	Count	*p*-Value
TNF signaling pathway	59	2.00 × 10^−7^
COVID-19	103	3.75 × 10^−7^
Epstein–Barr virus infection	92	4.31 × 10^−7^
Ribosome	73	4.12 × 10^−6^
IL-17 signaling pathway	46	6.00 × 10^−5^
Influenza A	73	1.06 × 10^−4^
Human cytomegalovirus infection	91	1.50 × 10^−4^

**Table 3 biomedicines-11-01531-t003:** The 5-fold cross-validation performance of drug-target prediction by our proposed DNN-based DTI model.

	Validation Loss	Validation Accuracy	Test Loss	Test Accuracy
1	0.196955	0.932880	0.132034	0.954051
2	0.222089	0.928476	0.118863	0.957865
3	0.186633	0.930830	0.122572	0.956865
4	0.196208	0.929585	0.119481	0.957610
5	0.207737	0.931498	0.109100	0.960423
Average	0.201924	0.930654	0.120410	0.957363
Standard Deviation	0.012096	0.001522	0.007361	0.002044

**Table 4 biomedicines-11-01531-t004:** Some candidate potential drugs predicted by the DNN-based DTI model for each biomarker we selected and their regulation ability, sensitivity, and toxicity.

Candidate Drugs	Regulation Ability (L1000)	Sensitivity (PRISM)	Toxicity (LC50, mol/kg)
Downregulation of IRF3			
acitretin	−0.057	−0.3305	2.328
erastin	−0.1905	−0.2612	1.532
RS-67333	−0.2883	0.0185	1.866
phenformin	−0.2192	NaN	2.261
Downregulation of STAT3			
acitretin	−0.0568	−0.3305	2.328
PRE-084	−0.0417	−0.2612	1.532
amiloride	−0.2368	0.1048	2.039
phenformin	−0.2541	NaN	2.261
Downregulation of TRAF6			
acitretin	−0.4937	−0.3305	2.328
PRE-084	−0.4749	−0.2612	1.532
amiloride	−0.2681	0.1048	2.039
phenformin	−0.4802	NaN	2.261
Downregulation of TYK2			
acitretin	−0.1998	−0.3305	2.328
erastin	−0.0572	−0.2612	1.532
RS-67333	−0.3243	0.0185	1.866
phenformin	−0.5475	NaN	2.261
Upregulation of MAVS			
phenformin	0.0091	NaN	2.261
megestrol-acetate	0.825	0.1951	1.902
remoxipride	0.426	−0.1732	2.015
RS-67333	0.520	0.0185	1.866

**Table 5 biomedicines-11-01531-t005:** The potential multimolecule drug screened by three drug design specifications from Table 4 for the therapeutic treatment of hRSV infection via a systematic drug-discovery approach.

Drug\Target	IRF3	STAT3	TRAF6	TYK2	MAVS
acitretin	**✓**	**✓**	**✓**	**✓**	
RS-67333	**✓**			**✓**	**✓**
phenformin	**✓**	**✓**	**✓**	**✓**	**✓**
Chemical structure of multimolecule drugs					
acitretin		RS-67333		phenformin	
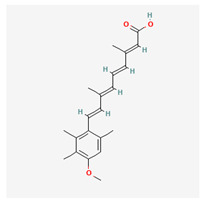		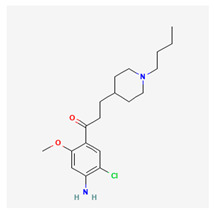		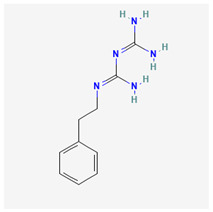	

## Data Availability

The RNA-seq datasets of hRSV infection are accessed from GSE55963 (https://www.ncbi.nlm.nih.gov/geo/query/acc.cgi?acc=GSE55963, accessed on 15 November 2022). The drug regulation ability data are from Phase I L1000 Level 5 datasets (https://www.ncbi.nlm.nih.gov/geo/query/acc.cgi?acc=GSE92742, accessed on 1 November 2021). The drug sensitivity datasets are from DepMapPRISM primary screen datasets (https://depmap.org/repurposing/, accessed on 1 November 2021). The source code is available on https://github.com/shusteven110/System-Biology-for-hRSV, accessed on 20 May 2023.

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
