# Peer review of "Genetic and Epigenetic Host–Virus Network to Investigate Pathogenesis and Identify Biomarkers for Drug Repurposing of Human Respiratory Syncytial Virus via Real-World Two-Side RNA-Seq Data: Systems Biology and Deep-Learning Approach"

_biomedicines, 2023, doi:10.3390/biomedicines11061531_

Round 1

Reviewer 1 Report

Title: Genetic and Epigenetic Host-Virus Network to Investigate Pathogenesis and Identify biomarkers for Drug Repurposing of Human Respiratory Syncytial Virus via Real-World Two-side RNA-Seq data: Systems Biology and Deep learning

The manuscript aims to determine biomarkers of RSV disease and identify a set of drugs that could be used to counteract the action of those biomarkers.  The authors use pathway data to construct a gene/protein – gene/protein interaction network using public databases and trim potential false positives using AIC approach. The authors use RNAseq time-course data from RSV infected cell lines to define a model of RSV infection over time and identify drug targets. Using Drug Target database data, the authors use a machine learning approach to drug targets for top genes associated with RSV disease. The authors identify a set of drugs that each target multiple of the top RSV genes. Overall, the authors demonstrate how high-quality time-course RNAseq data can be combined with existing database information used to identify multi-molecule targeting drugs computationally.

Comments

·       The major weakness of the manuscript is that the RSV infection model used to generate RNA seq data may not be appropriate. Heparin sulfate containing proteins are the prime interaction with G of RSV in cell lines, but is not present in the human lung and CX3CR1 appears to interact with G physiologically in the lung. The vero model may miss aspects of RSV infection in this regard and thus no targets could be identified that interfered with CX3CR1-G, an important aspect of RSV infection. This does not mean that the results are invalid, but may be limited and authors should discuss how their approach could be used with data from air-liquid interface/organoid in vitro models or in vivo models.

·       The specific RSV subtype (A or B) and strain (e.g. A2, Long, B1, … ) used in the in vitro model should be stated in the methods since different strains may interact differently with the host. Should be discussed that results may be subtype specific, especially in regards to G interactions.

·       It seems that Virus interactions were significantly decreased from Candidate GWGEN to Real GWGEN (table 1). Additionally, your core signaling pathway only includes 3 of the RSV gens (G, NS1, NS2). Was the lack of association with other RSV genes (e.g. M2-2) due trimming of the network or did these interaction not exist in the original data? Is the virus-host gene interactions more susceptible to your trimming approach than host-host genes?

·       For Figure 4, Triangle (RSV gene) not shown in Symbol table.

·       Palivizumab should be mentioned in introduction.

·       The biomarkers term is a little misleading in that the term often refers to gene/proteins associated with an illness, but not necessarily causative or highly relevant to the biological process. While I believe your approach is directly trying to identify biologically relevant gene/proteins.

Reviewer 2 Report

I congratulate the authors, the article is interesting and scientifically valid. I think it is very interesting given the high prevalence of RSV respiratory infections and the scarcity of specific drugs for its treatment. I have no particular remarks as the work, its organization and the presentation of the results are done in an excellent way. I have only a few minor observations.

1.       The authors in the Introduction section (lines 40 - 42) point out that ribavirin is the only drug approved by the FDA for the treatment of RSV. but that it is expensive and inconvenient. I would suggest adding a few more commentary words to explain why ribavirin is of little use in the treatment of RSV and what its side effects are (just one line).

2.       In the methods section (lines 153 - 154) the authors write that they used the human line A549 for HPI RNA-seq data. In my opinion the authors should spend a few lines explaining why they chose this cell line to obtain data for the present study. In addition, given that it is a lung cancer cell line, they should comment on the possibility that the data obtained are or are not fully attributable to those of a healthy epithelial cell. A few lines of comment are enough.

3.       In the results section (lines 518 - 529) the authors write that the viral infection increases the expression of CXCL8 and the activation of RELA, which in turn induces the expression of TNF which has a positive effect on the expression of IL- 6 and CXCL8 and this promotes inflammation and represses apoptosis favoring the virus. In the lines between 441 and 455 again it is emphasized that RSV infection increases the activation of FOXO3 (and RELA) and this increases the expression of IL-6 which in turn acts by activating STAT3 and causing its phosphorylation and passage at the nuclear level. I think it would be very important if, alongside the in silico analysis (moreover very well done) it would have been interesting if the authors had implemented in vitro experiments to confirm some of the observations made in silico. For example, the possibility of evaluating whether the increase in the expression of IL-8, IL-6 and TNF and the phosphorylation of STAT3 were observed on cells cultured in vitro and infected with RSV. By cytokine assays or RT-PCR assays, and for STAT3 phosphorylation by cytometric techniques. If it were possible to implement some similar experiment this would enrich the work.
